# Proteomics-based clustering outperforms clinical clustering in identifying people with heart failure with distinct outcomes
Marion van Vugt [1,2,3,4,16] ✉, Ruicong She[5,16], Isabella Kardys[6,16], Teun B. Petersen [6,7], Marie de Bakker [6], K. Martijn Akkerhuis[6], Kadir Caliskan [6], Olivier C. Manintveld[6], Alicia Uijl [2,3,8], Jan van Ramshorst[9], Dimitris Rizopoulos[7,10], Victor AWM Umans[9], Eric Boersma[6], David E. Lanfear[11,12,17], Folkert W. Asselbergs[2,3,13,14,17], Jessica van Setten [4,17] & A. Floriaan Schmidt [1,2,3,4,15,17] ✉

## Abstract

**Background** Heart failure (HF) clustering typically relies on clinical characteristics which may not reflect underlying pathophysiology relevant for personalized medicine. We aimed to identify plasma protein profiles of HF patients with reduced ejection fraction (HFrEF).
**Methods** Using latent class analysis, we derived clusters based on 1) clinical characteristics, and 2) proteomics (SomaScan) from 379 HFrEF patients (median age 64 years [Q1 56; Q3 72], 73% male). Survival analysis assessed associations with major cardiovascular (CV) events (HF hospitalization, CV death, or advanced therapy), HF hospitalization, CV death, and all-cause mortality. Associations were validated in 511 external patients (median age 72 years [Q1 63; Q3 79], 70% male). We identified differentially expressed proteins and explored whether proteins are targets of developmental or approved drugs.
**Results** We show that clinical clustering identifies three patient clusters without distinct disease progression. Contrary to this, clustering based on plasma proteomics identifies three patient clusters with clear differences in disease, which are validated in the external cohort. The slowly progressing cluster 1 includes younger patients with fewer comorbidities, while the rapidly progressing cluster 3 consists of older patients with more atrial fibrillation and renal failure, and the hospitalization cluster 2 is intermediate in many characteristics. Medication use is similar across clusters. Relative to cluster 1, patients in cluster 2 have an increased risk of major CV events (HR 2.31, 95%CI 1.23; 4.36) and HF hospitalization (HR 2.30, 95%CI 1.10; 4.78). Patients in cluster 3 experienced increased event rates of major CV events (HR 5.84), HF hospitalization (6.50), CV death (8.58), and all-cause mortality (5.07). Twelve proteins are differentially expressed across the identified clusters, including druggable CD2, GDF-15, ABO, IGFBP-1, IGFBP-2, and RNase1.
**Conclusions** Proteomics-based clustering identifies three HFrEF clusters associated with distinct outcomes that remain undetected using only clinical characteristics.

## Plain language summary

Heart failure affects millions of people worldwide, but symptoms and disease course varies greatly. People are often grouped based on basic clinical characteristics, which may miss important biological differences. In this study, we analyse blood proteins from people with heart failure and compare grouping based on these to a grouping based on clinical characteristics. We identify three biological groups of people with heart failure, and each group has a different future risk of hospitalization and death. The results are confirmed in an independent patient group. Our findings suggest that protein profiling can reveal hidden disease subtypes, which could help tailor treatments and improve outcomes for heart failure patients. We also identify proteins that could provide promising drug targets for specific patient groups.

Heart failure (HF) burdens society in terms of lost healthy life years, economic costs, and significantly reduces patients' quality of life. HF is a syndrome characterized by ventricular dysfunction, subclassified by the left ventricular ejection fraction (LVEF) as HF with reduced ejection fraction (HFrEF) for LVEF ≤ 40%, HF with mildly reduced ejection fraction for LVEF 41–49%, and HF with preserved ejection fraction for LVEF ≥ 50[1]. While this subclassification has clear clinical utility, the cutoff values for these categories are highly debated, and it is widely recognized that patients within the same HF subclass meaningfully differ in pathophysiology and aetiology[2]. The heterogeneity in HF patients is a potential cause of frequent and late-stage failure of drug trials for HF, where a subset of patients may benefit from treatment, but

the majority do not show sufficient benefit to warrant market authorization[3].

Previous studies have attempted to improve subclassification by combining clinical patient characteristics with unsupervised multivariate clustering to identify patients with a more homogenous presentation and indirectly increase the homogeneity of pathophysiology[4]. Proteins are fundamental elements in biological processes, and clustering based on proteins could therefore lead to subclasses with a similar underlying disease mechanism. For example, increased levels of the circulating N-terminal pro-B-type natriuretic peptide (NT-proBNP) are indicative of myocardial stretch[5], with NT-proBNP levels being used for cardiac diagnosis, prognosis, and therapy. Given that proteins are the targets of most drug compounds, clustering on plasma proteins may result in HF clusters with a more homogeneous response to treatment. Previous proteomics-based clustering analyses have sourced a limited number of 92 cardiovascular (CV) prioritized proteins[6]. Our previous investigation in a subset of the Bio-SHiFT study identified three clusters[6], whereas another study using the same 92 proteins identified six clusters[7]. Incorporating additional proteins that are not prioritized based on cardiovascular relevance may enhance clustering performance regarding disease association and may help identify novel proteins relevant for CV disease. In our previous work, we identified four clusters using a wide range of serially measured plasma proteins[8], whereas in the current study, we aim to develop an approach that could be more easily applied in clinical settings without requiring repeated measures. Finally, while previous studies have replicated findings by applying the same clustering method to different datasets, resulting in two or more models (i.e., a model for each dataset), they have not externally validated the same clustering model in independent data, hence limiting the clinical applicability of their results.

In the current study, we therefore directly compare an HFrEF clustering model based on clinical characteristics to a model based on 4210 plasma proteins combining latent class analysis (LCA) with principal component analyses. We include 379 HFrEF patients from the Bio-SHiFT study and externally validate the proteomic algorithm in 511 HFrEF patients from the Henry Ford HF PharmacoGenomic Registry (HFPGR). Subsequently, we explore differences in disease progression risk between HFrEF clusters and investigate druggability of the proteins driving cluster membership. Finally, to maximize applicability, we develop an open-access application programming interface allowing for the usage of our externally validated clustering model for other patients.

## Methods
### Study populations
Bio-SHiFT is a prospective cohort study of stable, chronic HF patients from Erasmus Medical Center Rotterdam, and Noordwest Ziekenhuisgroep, Alkmaar, the Netherlands[9]. Chronic HF patients aged 18 years or older were included during their regular outpatient visits. The study was approved by the Erasmus MC medical ethics committee, complied with the Declaration of Helsinki, and registered at ClinicalTrials.gov (NCT01851538). All patients provided written informed consent. At baseline, information was collected on HF symptoms and aetiology, CV risk factors, medical history, and medication use, and electrocardiography and echocardiography were performed. Baseline and three-monthly follow-up visits included short medical examinations and blood collection. A clinical event committee determined the outcomes based on hospital records and discharge letters, without knowledge of the proteomic measurements. The derived proteomics-based clustering model was externally validated in HFPGR, a prospective observational registry from Detroit, MI, USA, which included HF patients aged 18 years or older and diagnosed with HF as defined by the Framingham Heart Study[10,11]. Only HFrEF patients of European descent with SOMAscan data were included for validation. This study was approved by the Henry Ford Hospital review board and complied with the Declaration of Helsinki. All patients provided written informed consent at enrolment. In the current study, we considered (I) major CV event (a composite of HF hospitalization, CV death, heart transplantation, and left ventricular assist device implantation); (II) HF hospitalization; (III) CV death; and (IV) all-cause mortality. In BioSHiFT, HF hospitalization was defined as hospitalized for an exacerbation of HF symptoms, together with two of the following conditions: (a) brain natriuretic peptide or NT-proBNP more than three times the normal upper limit; (b) signs of worsening HF, including pulmonary rales, increased jugular venous pressure or peripheral oedema; (c) increased dose or intravenous administration of diuretics; or (d) administration of positive inotropic agents. Death was considered due to a CV cause if the cause of death included myocardial infarction, ischemic heart disease, heart disease, HF, sudden cardiac death, sudden undefined death, unwitnessed death, ill-described death, or stroke. In HFPGR, HF hospitalization was based on ICD9 and ICD10-codes, heart transplant and left ventricular assist device implantation on Current Procedural Terminology codes, and mortality was sourced from the Centers for Disease Control and Prevention—National Death Index (Supplementary Data 1).

### Plasma protein measurements
Blood samples of the BioSHiFT cohort were collected at baseline, processed within 2 h after collection, and stored at −80 °C for a median of 5.3 years (Q1 4.1; Q3 6.8). Proteomic analyses of ethylenediaminetetraacetic acid (EDTA) plasma samples were performed using the aptamer-based proteomic SOMAscan platform[12], and the quality control of the proteomics has been described previously[8,13]. In total, 4210 modified aptamers were measured, excluding aptamers with non-human, not validated targets, or with lower affinity than a competing aptamer for the same protein. In the current study, we used blood samples drawn at the time of inclusion and summarized the high-dimensional data using principal component analysis. The protein measurements were normalized to a mean of 0 and a standard deviation of 1; original values are reported in Supplementary Data 2. In the HFPGR validation cohort, blood samples were collected at enrolment, immediately aliquoted and stored at −70 °C for a median of 5.7 years (Q1 4.4; Q3 6.6). Proteomic analyses of EDTA plasma samples were performed using the SomaScan® V4 Assay, a platform for quantifying 5284 human proteins. The quality control was described previously[13,14], but in brief, systematic biases in raw assay data were corrected following SomaLogic data standardization protocols, involving multiple normalization and calibration steps. These included Hybridization Control Normalization, Intraplate Median Signal Normalization, and Median Signal Normalization to a global reference. Global reference standards were set for serum and plasma matrices, with controls, QC samples, and calibrators on each plate adjusted to these references. Any deviations in assay performance were monitored over time. An overall protein measurement quality metric was calculated for each sample, with all passing the recommended thresholds. The proteomics data were exported as SomaLogic ADAT files, which were imported into R using the *readat* package to remove these proteins with low quality.

### Statistics and reproducibility
We used LCA to derive a clustering model assigning 379 BioSHiFT HFrEF patients with similar clinical or proteomic profiles to clusters. In contrast to other clustering methods, LCA provides a straightforward function rule, which allows for assigning cluster membership to out-of-sample participants. For clinical clustering, variables with more than 5% missingness, related variables, and non-baseline variables arising during follow-up (such as drug prescriptions or devices) were excluded. Continuous clinical variables age, body mass index, and mean arterial pressure were categorized based on clinically relevant cut-offs, such as the World Health Organization classifications of normal weight, overweight, and obese for body mass index (Supplementary Data 3).

A genetic algorithm was employed to identify the subset of clinical variables used in LCA. In short, the genetic algorithm performs an exhaustive search for the most relevant clustering variables, comparing multiple subsets of variables simultaneously and by slightly mutating these subsets over many iterations[15,16]. For the clustering on plasma protein levels, categorization of the principal components (PCs) was done using different percentile-based cut-offs: quartiles (25th, 50th, 75th), deciles (10%

intervals), as well as custom groupings based on the 10th–90th and 20th–80th percentiles to define low, middle, and high values. Clustering was performed using all types of categorization, and the optimal categorization was chosen based on the highest entropy. The first 20 PCs were used, and a sensitivity analysis was performed using the first 30 PCs (Supplementary Fig. 1). Clusters were derived using maximum-likelihood estimation, with the optimal number of clusters identified using the Bootstrap Likelihood Ratio Test[17]. In short, we compared the $k$ cluster model with the lowest Bayesian Information Criteria with a $k + 1$ cluster model calculating the likelihood ratio for the two models. The distribution of the likelihood ratio was estimated using 999 bootstraps, selecting the model with $k + 1$ clusters based on a $p$-value of 0.05 or smaller. Patients were assigned to clusters based on the highest probability of cluster membership.

In addition to clinical and proteomic clustering, we performed a combined clustering analysis by integrating clinical clustering variables with the first 20 PCs derived from the proteomic variables. This approach aimed to determine if the combination of clinical and proteomic data improved patient stratification. The methodology was identical to that used for the clinical and proteomic clustering. As a sensitivity analysis, we performed the proteomic clustering applying latent profile analysis (LPA) using the R package *mclust*, which does not require categorization of the continuous principal components.

We next determined the contrasts between HFrEF clusters based on clinical characteristics (age, aetiology, and history of coronary artery disease [CAD], arrhythmia, hypertension, and smoking) and clusters based on the first 20 PCs of the plasma proteins. First, we evaluated the difference in baseline patient characteristics, followed by tests for the difference in the cumulative risk of a major CV event, HF hospitalization, CV death, and all-cause mortality, employing the log-rank test. Hazard ratios (HR) and 95% confidence intervals (CI) were calculated using Cox proportional hazards models, truncating follow-up at three years, and we calculated the c-statistic to assess the discriminative ability of these clusters.

### Annotation of differentially expressed proteins

To facilitate model explainability, we identified the differentially expressed proteins between the three proteomics-based clusters using the *limma* package in R. We corrected for multiple testing using the Benjamini–Hochberg procedure to control the false discovery rate while maintaining a balance between sensitivity and specificity. Proteins were considered significantly differentially expressed if they had a Benjamini–Hochberg adjusted $p$-value < 0.05 and a minimum absolute $\log_2$-fold-change of 1.0. Each protein was assigned to the cluster with the highest expression. To gain biological insight, we next explored pathway enrichment using the R package *clusterProfiler*, leveraging the gene ontology resource and applying a Benjamini–Hochberg adjusted $p$-value threshold of 0.05 on proteins with a minimum absolute $\log_2$-fold change of 0.7. Next, we benchmarked the differences in disease progression per cluster against the association of these individual proteins using a Cox regression model and presenting the protein-specific HRs (per standard deviation increase in protein value) and 95% CIs using the differentially expressed proteins, age, and sex. To further benchmark the cluster associations with disease progression, we compared these to the associations of canonical cardiovascular-related proteins alpha-actinin, BAG3, CRP, IL-6, TNF-alpha, cardiac troponin I, TnTc, tropomyosin, titin, and vinculin. Proteins were defined "druggable" if a drug targeting the protein is being tested in a clinical trial, and "drugged" if the compound has received marketing authorization[18]. ChEMBL[19] and the British National Formulary databases were queried to obtain information on the indications and side effects of compounds targeting these proteins.

### External validation of the proteomics-based clustering model

To validate our proteomics-based clustering model, we applied it directly to external data from HFPGR. Unlike prior studies that replicate their findings by applying the same clustering method to different datasets, resulting in

two or more clustering models, we derived a single clustering model in BioSHiFT to assign cluster membership externally. This approach aligns with potential clinical application, where clinicians aim to assign individual patients to clusters with distinct progression rates.

## Results

Bio-SHiFT includes 382 HFrEF patients of whom 379 have baseline SomaScan measurements available and are used in this study. Patients are included a median of 4.21 years [Q1 1.61; Q3 9.61] after their chronic HF diagnosis, they are predominantly male ($n = 275$, 73%), of European ethnicity ($n = 348$, 93%) with a median age of 64 years [Q1 56; Q3 72]. Determination of reduced LVEF and proteomics were obtained at baseline. Furthermore, 104 patients (28%) have a New York Heart Association (NYHA) class III/IV, 165 (48%) have a history of CAD, and the median NT-proBNP level is 1237 pg/mL [Q1 458; Q3 2463] (Supplementary Data 4).

### Deriving HFrEF patient clustering models

Leveraging clinical information on the 379 BioSHiFT patients results in a model assigning patients to three clusters with an ischemic aetiology, hypertensive aetiology, or underlying cardiomyopathy. The genetic algorithm flags six clinical characteristics (age, aetiology, and history of CAD, arrhythmia, hypertension, and smoking) as relevant for our multivariate clustering (Supplementary Fig. 2A, B). Cluster 1 consists for 85% of male patients, 94% of patients is diagnosed with CAD, 80% have suffered a myocardial infarction, 42% use antiplatelets, and 87% statins. The patients in cluster 2 are the oldest with a median age of 76 years, 88% of the patients is diagnosed with hypertension, 56% with atrial fibrillation, and 66% with renal failure. Cluster 3 consists of the youngest patients with a median age of 57 years, 69% of the patients have an underlying cardiomyopathy, and patients in this cluster have the lowest burden of comorbidities (Fig. 1, Supplementary Data 4). Using the plasma proteomics results in a clustering model which similarly assigns patients to three clusters (Supplementary Fig. 2C, D). The differences in patient characteristics are less evident compared to that of the clinical clusters. Proteomic cluster 1 consists of the youngest patients with a median age of 59 years, lower NYHA class (83% I/II), and a lower comorbidity burden. Patients in proteomic cluster 3 are older with a median age of 71 years, 61% have a history of CAD, 50% of atrial fibrillation, and 70% suffer from renal failure (Supplementary Data 5, Fig. 1). No differences are observed in medication use among the proteomic clusters. The patient overlap between the clinical and proteomic clusters is limited (Fig. 2).

Combined clustering, using the six clinical clustering variables and the first 20 PCs, results in three clusters (Supplementary Fig. 2E, F). Cluster 1 is almost identical to proteomic cluster 1, with a lower NYHA class (83% I/II) and a low comorbidity burden. Combined clusters 2 and 3 are very different from proteomic clusters 2 and 3. Combined cluster 2 consists of the oldest patients with a median age of 72 years and the highest comorbidity burden (Supplementary Data 6, Fig. 2).

The sensitivity analysis using LPA results in three clusters, with the first cluster consisting of younger individuals with a median age of 64 years and the highest percentage of females (32%). The second cluster groups 17 patients, with a median age of 72 years, 12 patients suffering from renal failure (71%) and six from diabetes (35%). The 125 patients in the third cluster have a median age of 65 years and the highest NYHA class (37% III/IV). Strikingly, the LVEF is equal among the three clusters (Supplementary Data 7).

### Associating cluster membership with outcomes

During a median follow-up of 2.24 [Q1 1.39; Q3 2.60] years, 113 patients experienced a major CV event, 89 patients were hospitalized for HF, 41 patients died, of which 32 suffered a CV death (Supplementary Data 4). Clinical cluster membership does not associate with the clinical outcomes and has a maximum c-statistic of 0.55 (95% CI 0.45; 0.64) for all-cause mortality. Compared to cluster 1, patients in combined cluster 2 and 3 have a very similar increased event rate for all outcomes except for all-cause

**Fig. 1 | Proportion of clinical characteristics per cluster.** Bars indicate the distribution of clinical characteristic for the clinical (left) and proteomic clustering (right). Numerical data underlying the figure are presented in Supplementary Data 4-5. AF atrial fibrillation, CAD coronary artery disease, MI myocardial infarction, NYHA New York Heart Association.

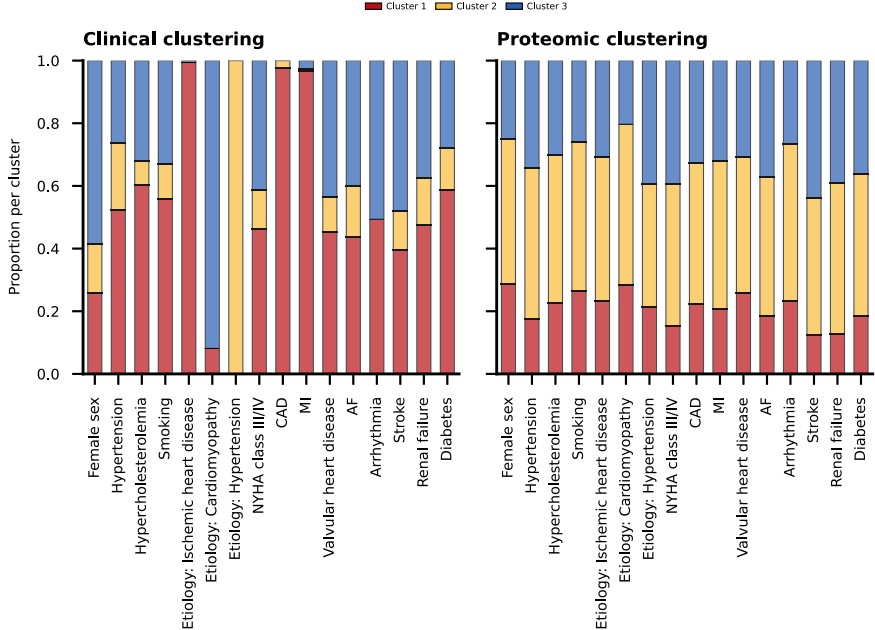

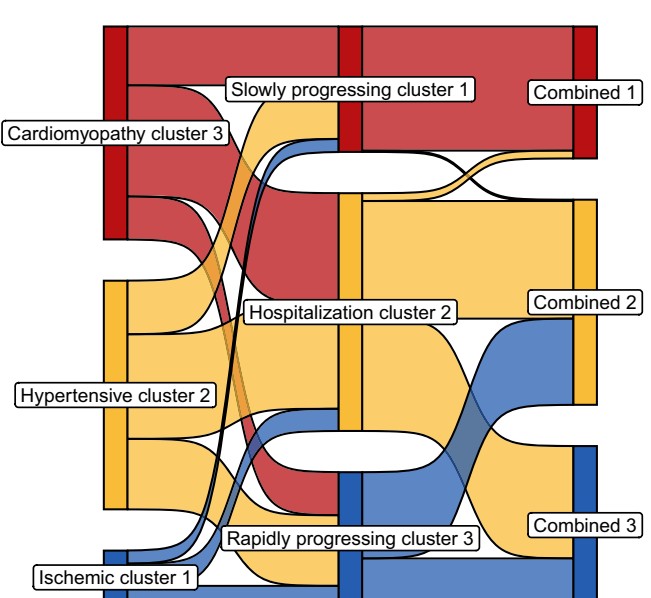

**Fig. 2 | HFrEF patient flow between the different clustering models.** The flow represents the proportion of HFrEF patients (*n* = 379) that were assigned to different clusters in the various approaches. For example, this figure shows that the clinical clusters (i.e., ischemic cluster 1 in blue, hypertensive cluster 2 in yellow, and cardiomyopathy cluster 3 in red) are very different from the proteomic clusters, whereas the proteomic slowly progressing cluster 1 and combined cluster 1 are almost identical.

a reference, patients in cluster 3 have an increased event rate for all outcomes, with limited difference comparing outcomes of cluster 2 to cluster 1. The c-statistic ranges between 0.58 and 0.63 for all outcomes (Supplementary Data 9).

Focussing on the disease associations of the proteomics-based clustering model, we find that compared to the slowly progressing cluster 1, patients in proteomics-based cluster 2 have an increased rate of major CV events (HR 2.31 95% CI 1.23; 4.36) and HF hospitalization (HR 2.30 95%CI 1.10; 4.78), which does not translate into an increased rate of fatal events (Fig. 4). Patients in the rapidly progressing proteomics-based cluster 3 have a more pronounced increased event rate compared to patients in slowly progressing cluster 1: HR 5.84 (95%CI 3.12; 10.93) for major CV events, and HR 6.50 (95%CI 3.17; 13.33) for HF hospitalization, which does result in an increased rate of fatal events: HR 8.58 (95%CI 2.56; 28.67) for CV death and HR 5.07 (95%CI 2.08; 12.33) for all-cause mortality (Fig. 4). The c-statistic for proteomics-based clustering ranges between 0.69 and 0.74 (Supplementary Data 8).

A sensitivity analysis based on the first 30 protein PCs similarly results in three clusters with comparable outcome associations, and LPA does not identify clusters with a similarly strong association with disease progression.

### Externally validating the proteomics-based clustering model
We next seek to externally validate the proteomics-based clustering model by replicating the associations between cluster membership and disease progression in 511 HFrEF patients from HFPGR. Patients were included a median of 6.03 years (Q1 1.98; Q3 10.87) after diagnosis with a median age of 72 years (Q1 63; Q3 79) and 69.9% are male (Supplementary Data 10). Here we find that, relative to the slowly progressing cluster 1, patients in cluster 2 have an increased rate of all-cause mortality (HR 2.37, 95%CI 1.37; 4.11). Similar to the derivation cohort, patients in rapidly progressing cluster 3 are at an increased risk of major CV events (HR 3.84, 95%CI 2.27; 6.50), HF hospitalization (HR 4.17, 95%CI 2.17; 8.00), CV death (HR 6.33, 95%CI 2.93; 13.64), and all-cause mortality (HR 6.65, 95%CI 3.87; 11.43; Fig. 4, Supplementary Data 11).

### Interpreting clustering model and identifying potentially druggable proteins
Next, to improve model interpretability, we identify differentially expressed proteins across the three proteomics-based derived clusters. We observe that ABO and tumor protein p53-inducible protein 11 (TP53I11) values are

mortality (Supplementary Fig. 3A, Supplementary Data 8). Protein cluster membership is significantly associated with major CV events, HF hospitalization, CV death, and all-cause mortality (Fig. 3, *p*-value < 0.001 for all outcomes). The disease associations of the combined clustering model (i.e., using both the clinical characteristics and protein values) are attenuated relative to the model exclusively using the available plasma proteins (Supplementary Fig. 3B, Supplementary Data 8). Combined cluster 2 is also associated with an increased risk of all-cause mortality (HR 3.15, 95% CI 1.19; 8.32). A consistently moderate c-statistic of ~0.61 is observed for all outcomes (Supplementary Data 8). Taking cluster 1 identified by the LPA as

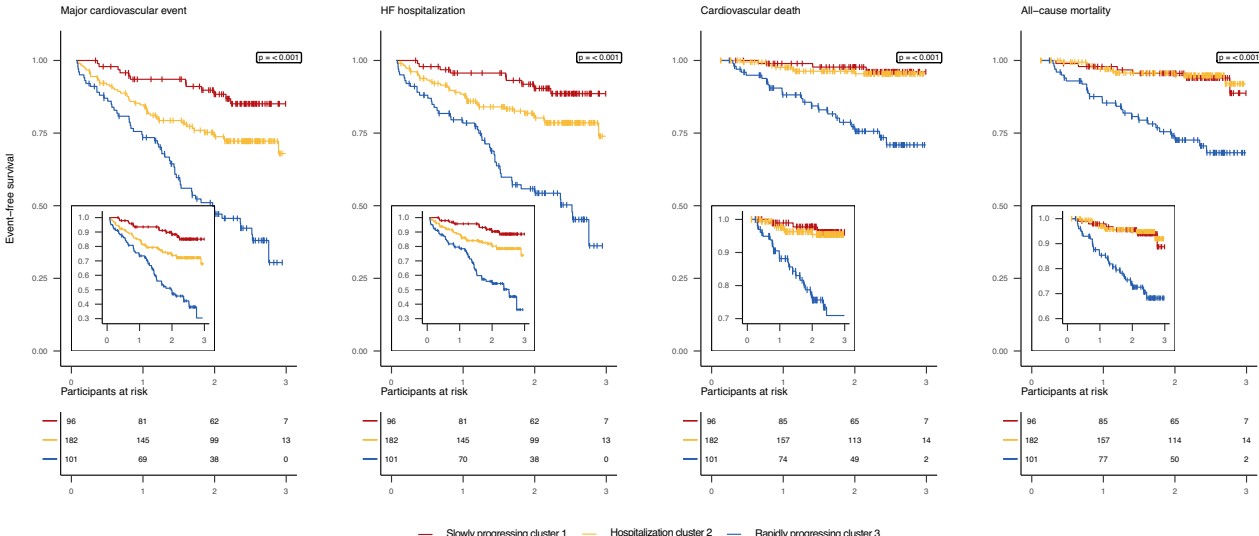

**Fig. 3 | Event-free survival is significantly different across proteomics-based clusters.** Kaplan–Meier curve for the clinical outcomes stratified by proteomic clusters. Differences were assessed using the two-sided log-rank test without adjustment for multiple testing, resulting in significant differences for all outcomes: $p = 7.57 \times 10^{-11}$ for major cardiovascular event, $p = 2.91 \times 10^{-10}$ for HF hospitalization, $p = 1.53 \times 10^{-8}$ for cardiovascular death, and $p = 2.00 \times 10^{-8}$ for all-cause mortality. HF heart failure.

**Fig. 4 | The association between proteomics-based cluster membership and disease progression in people with HFrEF.** Follow-up was truncated at three years. The model was derived in 379 participants of the BioSHiFT cohort and externally validated in 511 participants of the HFPGR. Bars represent the 95% confidence interval. Numerical data underlying the figure are presented in Supplementary Data 9. CI confidence interval, HF heart failure, HFPGR Henry Ford HF PharmacoGenomic Registry, HR hazard ratio.

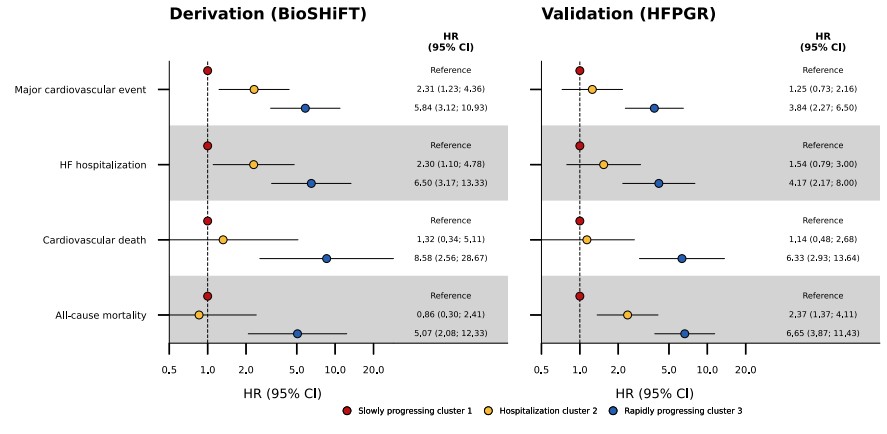

significantly upregulated in the slowly progressing cluster 1, whereas the values of ten proteins are significantly upregulated in the rapidly progressing cluster 3 (showing intermediate high values in hospitalization cluster 2): peroxidasin homolog (PXDN), neuroblastoma suppressor of tumorigenicity 1 (NBL1), RNase1, insulin-like growth factor binding protein (IGFBP) 2, REG-3-alpha, growth-differentiation factor-15 (GDF-15), Apolipoprotein F, NT-proBNP, IGFBP-1, and CD2 (Supplementary Data 12). Filtering for a $\log_2$-fold-change of 0.7 results in 56 additional proteins, and pathway enrichment analysis shows that slowly progressing cluster 1 is characterized by proteins involved in lipid metabolism (such as ADH4, APOA5, APOC3, and CRYZL1). Rapidly progressing cluster 3 is characterized by proteins involved in the immune response (such as EPHA2, RNase1, MMP12, NBL1, and REG-3-alpha) and extracellular matrix organisation (such as COL28A1, CST3, PRSS2, PXDN, and TNFRSF1A). Hospitalization cluster 2 is characterized by energy metabolism and glucose usage (Supplementary Data 13).

To benchmark cluster-outcome associations, we compare the HRs (Fig. 4) to the disease HRs of the differentially expressed protein values, supplemented by the HRs of ten canonical proteins (Fig. 5, Supplementary Data 14). While NT-proBNP, IGFBP-2, GDF-15, NBL1, and PXDN are

significantly associated with clinical outcomes, their individual associations are markedly smaller than the HRs for cluster membership (Figs. 4 and 5, Supplementary Data 14).

Of the twelve differentially expressed proteins, CD2 is targeted by drug compounds alefacept and siplizumab, which are indicated for diabetes type I, plaque psoriasis, and arthritis. Additionally, GDF-15, ABO, IGFBP-1, IGFBP-2, and RNase1 are druggable. GDF-15 is inhibited by visugromab and ponsegromab which are evaluated in clinical phase oncological trials and ponsegromab for heart failure (Supplementary Data 15).

## Discussion

In the current study, we find that a plasma protein-based model derives three HFrEF clusters with distinct outcome associations. In contrast, a clustering model based on clinical variables resulted in aetiological clusters which does not associate with HFrEF outcomes. Furthermore, after externally validating our proteomics-based clustering model, we have made this publicly available for potential usage in clinical settings: https://gitlab.com/mvvugt/clusimp.

Patients in proteomics-based rapidly progressing cluster 3 have worse outcomes for all clinical outcomes compared to slowly progressing cluster 1

**Fig. 5 | Association of differentially expressed proteins and canonical cardiac proteins with clinical outcomes.** Follow-up was truncated at three years. The model was derived in 379 participants of the BioSHiFT cohort. Bars represent the 95% confidence interval. CI confidence interval, HF heart failure, HR hazard ratio.

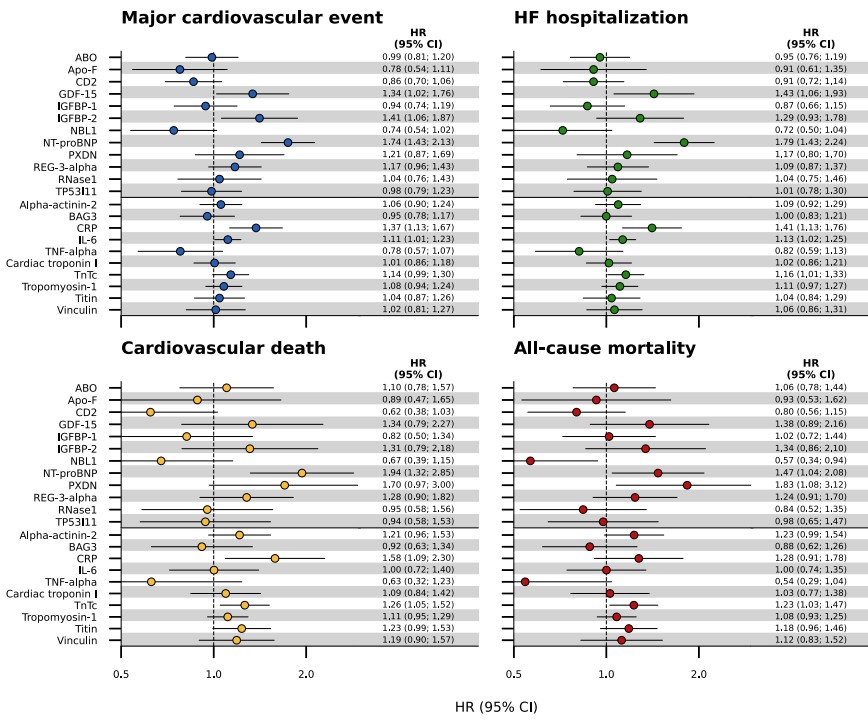

and hospitalization cluster 2, which is also observed in the external validation cohort. Interestingly, patients in hospitalization cluster 2 have worse outcomes for major CV events and HF hospitalization compared to those in slowly progressing cluster 1 but show a comparable low rate of fatal events compared to slowly progressing cluster 1. Despite a strong association with disease progression, medication use is equal in all proteomics-based clusters, implying that patients are treated suboptimally and indicating the potential for personalized treatment based on their proteomics profile.

Out of the twelve differentially expressed proteins, eight have been previously associated with cardiac traits or diseases. Specifically, common genetic variants in the *ABO* gene have been associated with heart failure[20–23] and myocardial infarction[24–26], while variants in *TP53I11* have been associated with resting heart rate[27,28] and hypertrophic cardiomyopathy[29]. These proteins are significantly upregulated in slowly progressing cluster 1, which is characterized by an altered lipid homeostasis. Hospitalization cluster 2 is intermediate in many aspects, such as expression of the proteins, and characterized by glucose usage in the pathway analysis.

PXDN has previously been shown to be upregulated in HFrEF patients compared to healthy individuals[30], where it plays a role in fibrosis and extracellular matrix formation. RNase1 has been proposed as HF biomarker due to its increased expression in HF patients and cardioprotective effects by catalysing RNA degradation, thereby mitigating inflammatory responses[31]. RNase1, NBL1, and REG-3-alpha are involved in immune response mechanisms and together with PXDN, these proteins and processes characterize the rapidly progressing cluster 3. Both IGFBP-1 and IGFBP-2 are involved in inflammation and metabolism, and modulate the effects of IGF-1, which is crucial for cellular development and survival. They are hypothesized to play a role in HF pathophysiology and have been suggested as biomarkers for mortality and CV disease risk[32–35]. The combination of high levels of IGFBP-1 and NT-proBNP has previously been associated with worse prognosis in HFrEF patients[7]. GDF-15 is a stress response protein highly expressed in cardiomyocytes upon several HF-related pathophysiological conditions such as inflammation, oxidative stress, hypoxia, and tissue injury. Compared to NT-proBNP, another stress biomarker in HF, GDF-15 is expressed in more tissues and therefore provides different information compared to NT-proBNP in a systemic disease such as HFrEF[36]. Both proteins have been associated with HF severity and prognosis[37–42], with GDF-15 proposed as an HF drug target. Drugs are currently being developed for GDF-15, offering potential opportunities for repurposing, while other proteins such as ABO, IGFBP-1, IGFBP-2, and RNase1 present promising targets due to their druggability.

Benchmarking the association between cluster membership with disease progression against the individual associations of the differentially expressed proteins shows a substantially stronger association than for any individual protein (HR between 5.07 and 8.59 for rapidly progressing cluster 3, compared to an HR < 2 for any of the individual proteins). This includes higher values of NT-proBNP and GDF-15, which show an HR per standard deviation of at most 2. This suggests that the joint consideration of multiple protein values by the clustering model is essential to more accurately identify people with a meaningfully worse disease outcome.

We anticipate that significant survival differences for clinical clustering and the attenuated association for combined clustering may become evident in larger sample size settings. For example, a clustering study on 6909 HF patients with preserved ejection fraction applying similar methods, identified five clusters based on clinical characteristics which were associated with disease outcomes[43]. Hence, our results do not show an absence of differences between patients clustered using clinical characteristics or clinical characteristics combined with proteins but rather suggest that the difference is larger when using proteomics exclusively. Despite the improvement compared to clinical clustering, the c-statistics and high-dimensional nature of the proteomic clustering limits immediate clinical applicability. Our findings represent a first step toward identifying biologically distinct HFrEF subgroups. Future work should aim to develop a minimal, robust set of proteins that can more accurately assign patients to clusters. These steps are not pursued in the present study to avoid overfitting due to repeated analyses on the same data. Future studies should also explore the contribution of the over-expressed proteins to cardiac metabolism and pathogenesis, adding to the known associations with cardiac disease. These proteins might also be useful in other clinical settings, for which clinically useful cut-off values should be established to guide individual patient management. However, in our study, cut-off values are not required because the clustering method assigns patients to clusters based on their overall proteomic profile. Our findings are based on proteomic data from one timepoint, a median of 4.21 years after initial diagnosis. Protein values are anticipated to be influenced by disease stage and concomitant

medication usage at the time of proteomics measurement has potential implications for the derived clustering model. Some variation may reflect this biological and clinical heterogeneity, which is also representative of real-world clinical settings. Medication usage is similar across the clusters, and thus unlikely to have caused the observed differences in associations with clinical outcomes. Importantly, we have externally validated the clustering model in a US healthcare setting where patients were recruited a median of 6.03 years after initial diagnosis. The storage duration of the blood samples may have affected the stability of the proteins. We expect the influence on our results to be limited because storage duration is relatively consistent across patients. While protein degradation might have affected the detectability of individuals proteins and the set of differentially expressed proteins, the clustering approach is likely robust to this variation, because it is based on relative patterns across the proteome. Nevertheless, application of the clustering model in newly diagnosed HF patients deserves further careful consideration. This study, as well as previous clustering studies, includes predominantly male HF patients from European descent. While this reflects the higher risk of HFrEF observed in males[44–46], this imbalance requires careful consideration regarding generalizability. To enhance the clinical applicability of our work, we limited the study to HFrEF patients and to support similar studies in HF with preserved ejection fraction, mildly reduced ejection fraction, or any HF population, we have released the computational code. Due to the moderate sample size, we categorize continuous variables to simplify the model and reduce sample size requirements, which might have affected model expressivity. In larger sample size settings, more flexible algorithms may further improve the already highly discriminative model's current performance. Nevertheless, the derived model is successfully validated in an external cohort that differs considerably from the derivation cohort, underscoring the robustness of our approach.

## Conclusion

We derive and externally validate a clustering model leveraging information on plasma proteins to identify three groups of HFrEF patients associated with distinct disease progression. These differences in disease progression are more pronounced when patients are clustered on clinical characteristic only. Cluster membership is driven by twelve differentially expressed proteins, of which six are drugged or druggable, providing potential leads for HF drug development.

## Data availability

The clustering algorithm is available online at https://gitlab.com/mvvugt/clusimp. Anonymized data that support the findings of this study will be made available to other researchers for purposes of reproducing the results upon reasonable request and in accordance with a data-sharing agreement. Requests can be directed to I. Kardys (i.kardys@erasmusmc.nl). The source data for Fig. 1 is in Supplementary Data 4, for Fig. 3 and Supplementary Fig. 3 in Supplementary Data 8, for Fig. 4 in Supplementary Data 10, and for Fig. 5 in Supplementary Data 14.

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

## Acknowledgements

The Bio-SHiFT study was supported by the Jaap Schouten Foundation. HFPGR was funded in part by NIH grants (R01HL103871 and R01HL132154; D.E.L.). D.E.L.'s time is supported in part by grants from the NIH (P50MD017351 and R21HL168695). Moreover, this work was funded by UK Research and Innovation (UKRI) under the UK government's Horizon Europe funding guarantee EP/Z000211/1 [A.F.S.]; the Rosetrees CF-2-2023-M-2/122 [A.F.S.]; the Dutch Heart Foundation [grant number 2019T045, to M.v.V. and J.v.S.]; the EU/EFPIA Innovative Medicines Initiative 2 Joint Undertaking BigData@Heart [grant number 116074, to F.W.A.]; the British Heart Foundation [grant number PG/22/10989, to A.F.S.]; the UCL BHF Research Accelerator [grant number AA/18/6/34223, to A.F.S.]; and the National Institute for Health and Care Research University College London Hospitals Biomedical Research Centre to A.F.S. This publication is part of the project Computational medicine for cardiac disease with file number 2025.027 of the research programme Round Computing Time National Computer Facilities which is (partly) financed by the Dutch Research Council (NWO) under the grant ID https://doi.org/10.61686/WBEKI87000. The authors acknowledge the use of the UCL Myriad High Performance Computing Facility (Myriad@UCL), and associated support services, in the completion of this work.

## Author contributions

Study conception: A.F.S. Manuscript drafting: M.v.V.; Data acquisition: R.S., I.K., T.B.P., M.d.B., K.M.A., K.C., O.C.M., J.v.R., V.A.W.M.U., D.E.L.; Study design BioSHiFT: I.K., K.M.A., V.A.W.M.U., E.B.; Study design and data analysis: M.v.V., R.S., A.U., A.F.S.; Project supervision and funding acquisition: I.K., D.R., E.B., D.E.L., F.W.A., J.v.S., A.F.S.; All authors reviewed and approved the final version.

## Competing interests

O.C.M. has served on advisory boards of Abbott, Astra Zeneca, Boehringer Ingelheim, and Novartis. A.F.S. has received funding from New Amsterdam and Servier for unrelated work and is an Editorial Board Member for Communications Medicine but was not involved in the editorial review or peer review, nor in the decision to publish this article. I.K. has received travel reimbursement from SomaLogic and Olink. D.E.L. is a consultant for Astra Zeneca, Bayer, Cytokinetics, Illumina, RyCarma, has participated in research with Akros, AstraZeneca, Cytokinetics, Lilly, Kardigan, Novartis, Pfizer, Somalogic, and has a patent (held by Henry Ford Health) for a beta-blocker response polygenic score. All other authors declare no competing interest.

## Additional information

[1]Institute of Cardiovascular Science, Faculty of Population Health, University College London, London, UK. [2]Department of Cardiology, Amsterdam Cardiovascular Sciences, Amsterdam University Medical Centre, University of Amsterdam, Amsterdam, The Netherlands. [3]Amsterdam Cardiovascular Sciences, Heart Failure and Arrhythmias, Amsterdam, The Netherlands. [4]Division Heart & Lungs, Department of Cardiology, University Medical Center Utrecht, Utrecht University, Utrecht, The Netherlands. [5]Department of Public Health Sciences, Henry Ford Hospital, Detroit, MI, USA. [6]Department of Cardiology, Thorax Center, Cardiovascular Institute, Erasmus MC, University Medical Center Rotterdam, Rotterdam, The Netherlands. [7]Department of Biostatistics, Erasmus MC, University Medical Center Rotterdam, Rotterdam, The Netherlands. [8]Department of Clinical Science and Education, Södersjukhuset, Karolinska Institutet, Stockholm, Sweden. [9]Department of Cardiology, Northwest Clinics, Alkmaar, The Netherlands. [10]Department of Epidemiology, Erasmus MC, University Medical Center Rotterdam, Rotterdam, The Netherlands. [11]Department of Internal Medicine, Center for Individualized and Genomic Medicine Research, Henry Ford Hospital, Detroit, MI, USA. [12]Division Cardiovascular Medicine, Department of Medicine, Henry Ford Hospital, Detroit, MI, USA. [13]Institute of Health Informatics, University College London, London, UK. [14]National Institute for Health Research, University College London Hospitals, Biomedical Research Centre, University College London, London, UK. [15]UCL British Heart Foundation Research Accelerator, London, UK. [16]These authors contributed equally: Marion van Vugt, Ruicong She, Isabella Kardys. [17]These authors jointly supervised this work: David E Lanfear, Folkert W Asselbergs, Jessica van Setten, A. Floriaan Schmidt. ✉e-mail: m.vugt@ucl.ac.uk; amand.schmidt@ucl.ac.uk

