## [Transparent Peer Review file · Communications Medicine]

Proteomics-based clustering outperforms clinical clustering in identifying people with heart failure with distinct outcomes

Corresponding Author: Dr Marion van Vugt

Version 0:

Reviewer comments:

Reviewer #1

(Remarks to the Author)

In this study, Vugt et al. compared the performance of proteomics-based clustering and clinical-based clustering in identifying adverse events in HFrEF patients. Overall, this study sheds new light on HFrEF management and new drug development. However, enthusiasm is dampened by the following, which need to be addressed:

1. The author did not specify the timing of phlebotomy among these patients, especially the time between the HFrEF's first diagnosis and the phlebotomy. The level of these proteins may vary in different disease periods. Would medication affect the proteins levels? Moreover, was HFrEF was the first diagnosis of these patients? As we all know the LVEF might be changed, up or down. For example, patients with preserved EF or mildly reduced EF could be involved into reduced EF if the self-management was poor. This would also be worth discussing. Please comment on that.
2. In this manuscript, the author implies the significance of the Proteomics-based clustering model in clinical practice. However, the author did not assess it. A decision curve analysis would be helpful in this situation. I want the author to complement it in the manuscript.
3. The author mentioned that in the models the continuous variables were turned into categories, however, the detailed information on which variables were not supplied. Please comment on that.
4. Though the Proteomics was promising, was it enough easy to get and economic? We have to acknowledge that patients management need to balance the individualize and well applicability. Please comment on that.
5. A cut-off value of the Proteomics-related biomarkers would make it easier to use in clinical practice.

Reviewer #2

(Remarks to the Author)

The authors present a substantial amount of work combining SomaScan proteomics profiling to explore clustering of HFrEF patients, and performing validation in an independent cohort. The scope of statistical analyses performed and the inclusion of external validation are commendable. There are however some point that should be clarified in the manuscript.

Specific points:

1. The manuscript lacks a description of the duration of plasma storage prior to SomaScan analysis. It is well established that the stability of plasma proteins can degrade during long-term storage, especially for low-abundant proteins - which are often the targets of the SomaScan platform. The authors should specify the storage duration to better assess the reliability of the analyses.
2. There is a discrepancy between the Methods section and the supplementary methods regarding the temperature at which the plasma samples were stored. The manuscripts state -80°C (p.7; s. 13), whereas the supplementary methods state -70°C (p.4; s.5). Please clarify which is correct.
3. Also in the supplementary methods, the description (p.4; s.16) alludes to additional quality checks and filtering. For transparency and reproducibility, the authors should describe these processes in more detail, including the filtering criteria applied.
4. Similarly, on p.8, s.12-13, the authors mention that correction for the number of independent tests was performed. It is

important to specify which method was used for correction for multiple testing and please justify the choice.

5. The authors should also clarify how missing values were handled for the proteomic data, and whether the proteomic analyses were blinded to the clinical outcome.

6. While the study is validated in a secondary cohort, the reported c-statistics remain modest, with values in the range of 0.69-0.74 for the derivation cohort and 0.65-0.71 for the validation cohort. Such performance is generally considered of limited clinical utility. Although the benchmarking show improvement over clinical characteristic alone, the clinical relevance of the findings still eludes me. I will encourage the authors to discuss these limitations more explicitly and consider the added benefit for translation into clinical practice.

Reviewer #3

(Remarks to the Author)

This study showed that Proteomics-based HFrEF clustering identified three clusters associated with distinct disease outcomes, which were not detected using clinical characteristics. However, the statistical analysis and related Figures (Tables) seems insufficient in main text. The methods description such as the QC and important variables were not described in detail. Despite the author constructed a validated clustering algorithm, considering the SomaScan proteomic include more than 4000 proteins, it is hard to replicate the clustering due to the cost. The author conducted the differential expression analysis to identify 12 markers, however there is no further analysis to test the accuracy and efficiency of using these 12 markers for phenotyping, nor conducting any other approaches for interpretation and application as sensitivity analysis. These limitations may influence the explanation and clinical implication of this study.

1. The definition of "major cardiovascular (CV) events" should be illustrated in Abstract.
2. I recommend the application interface address can be described in the "Results" section but not in the Abstract and Introduction.
3. "Clustering on clinical characteristics identified three patient clusters that did not differ in disease progression rates". What does the clinical characteristics contain? These information about the input variables should be included in Methods section.
4. The Quality Control Process and Results of Quality (such as important QC statistics and QC Figures) should be included in main text (It is better to demonstrate a summary of QC in main text and to illustrate in detail in the Supplementary Material).
5. "A sensitivity analysis based on the first 30 protein PCs similarly resulted in three clusters with comparable outcome associations.". How are the first 30 proteins determined?
6. How does the statistical significance of differential expressed proteins test? Such as the cut-off of P-value? And is the comparison conducted in 3 clusters to report P for trend, or other comparison methods?
7. The Figure Legend and Figure explanation of Figure 1 were missing.
8. The description for characteristics of each cluster should be better demonstrate in both text and figure (table). I recommend a summary of each cluster's characteristic should also be included in Abstract.

Reviewer #4

(Remarks to the Author)

This manuscript describes the use of proteomics data generated using SomaScan to discover latent classes. The main aim is to identify plasma protein profiles of HF patients with reduced ejection fraction (HFrEF).

1. The choice to discretize PCA scores into quartiles seems arbitrary. It would be helpful for the authors to justify why quartiles were selected, and to clarify whether different categorization schemes were evaluated for robustness (i.e., whether the final clustering results depend heavily on how the PCA scores were divided).
2. It's not very clear whether the latent classes reflect actual patient subgroups or are potentially artifacts of preprocessing choices due to the use of PCA to categorize the proteomics data.
3. Did the authors consider using LPA instead of LCA for the proteomics data?
4. Have the authors consider using pathway enrichment or gene set enrichment analysis (e.g., GSEA) within each cluster to better understand the molecular processes or pathways underlying the observed clusters?
5. While using \log_2 fold change >1 is reasonable, the authors may consider exploring a lower cutoff (e.g., 0.5) to capture more subtle shifts between clusters.

Version 1:

Reviewer comments:

Reviewer #1

(Remarks to the Author)

The author answered my comments well and discussed the limitations properly. I had no further comments. Thank you.

Reviewer #2

(Remarks to the Author)

I have no further comments to the manuscript. The authors have adequately addressed my previous comments.

Reviewer #3

(Remarks to the Author)

The authors have addressed and revised most of my comments. I still have some minor suggestions regarding the tables and figures in the manuscript. If there is no restriction on the number of tables and figures, I would still recommend increasing their quantity—for example, by moving Figure S4 and S5 into the main text. As the authors highlighted in the abstract: "Twelve proteins were differentially expressed, including druggable targets CD2, GDF-15, ABO, IGFBP-1, IGFBP-2, and RNase1." It is also advisable to add a table listing the drug-target profiles of these key proteins.

Reviewer #4

(Remarks to the Author)

Many thanks to the authors for their response. I have no further comments.

Response to the reviewers

Referee #1

In this study, Vugt et al. compared the performance of proteomics-based clustering and clinical-based clustering in identifying adverse events in HFrEF patients. Overall, this study sheds new light on HFrEF management and new drug development. However, enthusiasm is dampened by the following, which need to be addressed:

1. The author did not specify the timing of phlebotomy among these patients, especially the time between the HFrEF's first diagnosis and the phlebotomy. The level of these proteins may vary in different disease periods. Would medication affect the proteins levels? Moreover, was HFrEF was the first diagnosis of these patients? As we all know the LVEF might be changed, up or down. For example, patients with preserved EF or mildly reduced EF could be involved into reduced EF if the self-management was poor. This would also be worth discussing. Please comment on that.

Response: The reviewer raises important points regarding the timing of phlebotomy, the potential influence of medication on protein levels, and the dynamic nature of LVEF over time. To address the variation of protein levels, we have now included a more elaborate discussion on these factors in the discussion. Specifically, we report the median time between first HF diagnosis and baseline assessment (moment of blood draw). We furthermore clarify that the phlebotomy and determination of LVEF were performed at the same time, eliminating the concerns regarding evolved EF in the period between LVEF determination and proteomic measurement.

The information on periods between HFrEF diagnosis, phlebotomy, and LVEF determination is included in the Results section on page 11 (lines 6-13):

“Bio-SHIFT included 382 HFrEF patients of whom 379 had baseline SomaScan measurements available and were used in this study. Patients were included a median of 4.21 years [Q1 1.61; Q3 9.61] after their chronic HF diagnosis, they were predominantly male (n=275, 73%), of European ethnicity (n=348, 93%) with a median age of 64 years [Q1 56; Q3 72]. Reduced LVEF and proteomics were obtained at baseline. Furthermore, 104 patients (28%) had a New York heart association (NYHA) class III/IV, 165 (48%) had a history of coronary artery disease, and the median NT-proBNP level was 1,237 pg/mL [Q1 458; Q3 2,463] (Table 1, Table S4).”

The potential change of protein values across disease course is addressed from page 17 line 14 until page 18 line 2:

“Our findings are based on proteomic data from one timepoint, a median of 4.21 years after initial diagnosis. Protein values are anticipated to be influenced by disease stage and concomitant medication usage at the time of proteomics measurement has potential implications for the derived clustering model. Some variation may reflect this biological and clinical heterogeneity, which is also representative of real-world clinical settings. Medication usage was broadly similar across the clusters, and thus unlikely to have caused the observed differences in associations with clinical outcomes. Importantly, we have externally validated the clustering model in a US healthcare setting where patients were recruited a median of 6.03 years after initial diagnosis. The storage duration of the blood samples may have affected the stability of the proteins. We expect the influence on our results to be limited, because storage duration was relatively consistent across patients. While protein degradation might have affected the detectability of individuals proteins and the set of differentially expressed proteins, the clustering approach is likely robust to this variation, because it is based on relative patterns across the proteome. Nevertheless, application of the clustering model in newly diagnoses HF patients deserves further careful consideration.”

2. In this manuscript, the author implies the significance of the Proteomics-based clustering model in clinical practice. However, the author did not assess it. A decision curve analysis would be helpful in this situation. I want the author to complement it in the manuscript.

Response: We thank the reviewer for this suggestion and have included a decision curve analysis (DCA) in Figure R1. This demonstrates that stratified clinical decision-making based on our clustering solution generally yields greater net benefit than a treat-all approach. We would like to note, however, that DCA is typically applied to probabilistic risk prediction models that produce continuous estimates between 0 and 1. In contrast, our clustering approach defines three discrete subgroups (one per cluster), each associated with a single empirical outcome rate. Consequently, the DCA presented is based on only three fixed risk strata. We are concerned that including this figure in the main manuscript might inadvertently suggest that our model supports continuous individualised risk prediction, which it does not.

Figure R1. Net benefit per outcome for the proteomics-based clustering.

3. The author mentioned that in the models the continuous variables were turned into categories, however, the detailed information on which variables were not supplied. Please comment on that.

Response: We have moved the information on the categorised continuous variables to the main text of the manuscript and included some additional details on which variables were categorised. For the exact categorisation we refer to Table S3, which is repeated below as well.

The methods section on page 9 (lines 2-8) now reads:

“For clinical clustering, variables with more than 5% missingness, related variables, and non-baseline variables arising during follow-up (such as drug prescriptions or devices) were excluded. Continuous clinical variables age, body mass index, and mean arterial pressure were categorized based on clinically relevant cut-offs, such as the World Health Organization classifications of normal weight, overweight, and obese for body mass index (Table S3), and for the plasma proteins, the principal components (PCs) were categorized using quartiles.”

Table S3. Cut-offs for categorized clinical variables

Variable	Category	Values
Age	<65	<65
	65-80	65-80
	>80	>80
BMI	<25	<25
	25-30	25-30
	>30	>30
MAP	Low	<70
	Normal	70-110
	High	>110
Abbreviations: BMI = body mass index,		
MAP = mean arterial pressure		

4. Though the Proteomics was promising, was it enough easy to get and economic? We have to acknowledge that patients management need to balance the individualize and well applicability. Please comment on that.

Response: We fully agree and have provided further discussion that our proof-of-principle study requires not only additional validation in people with a recent HF diagnosis, but that the algorithm may also be further optimised in terms of costs by reducing the set of proteins. Please see page 17, lines 3-9:

“Despite the improvement compared to clinical clustering, the c-statistics and high dimensional nature of the proteomic clustering limits immediate clinical applicability. Our findings represent a first step toward identifying biologically distinct HFrEF subgroups. Future work should aim to develop a minimal, robust set of proteins that can more accurately assign patients to clusters. These steps were not pursued in the present study to avoid overfitting due to repeated analyses on the same data.”

5. A cut-off value of the Proteomics-related biomarkers would make it easier to use in clinical practice.

Response: We fully agree that cut-off values or reference ranges are important when evaluating measurements of individual proteins. The presented work expands on this *modus operandi* by identifying and validating clusters characterised by multiple proteins, applying data-driven cut-offs not to differentiate between normal and abnormal but simply to group

people in distinct groups based on their protein value. We have clarified this on page 17 lines 9-14:

“Future studies should also explore the contribution of the overexpressed proteins to cardiac metabolism and pathogenesis, adding to the known associations with cardiac disease. These proteins might also be useful in other clinical settings, for which clinically useful cut-off values should be established to guide individual patient management. However, in our study, cut-off values are not required because the clustering method assigns patients to clusters based on their overall proteomic profile.”

Referee #2

The authors present a substantial amount of work combining SomaScan proteomics profiling to explore clustering of HFREF patients, and performing validation in an independent cohort. The scope of statistical analyses performed and the inclusion of external validation are commendable. There are however some point that should be clarified in the manuscript.

Specific points:

1. The manuscript lacks a description of the duration of plasma storage prior to SomaScan analysis. It is well established that the stability of plasma proteins can degrade during long-term storage, especially for low-abundant proteins - which are often the targets of the SomaScan platform. The authors should specify the storage duration to better assess the reliability of the analyses.

Response: We have reported the blood sample storage duration in the methods section for both cohorts. Please see page 8, lines 12-13:

“Blood samples of the BioSHiFT cohort were collected at baseline, processed within two hours after collection, and stored at -80°C for a median of 5.3 years (Q1 4.1; Q3 6.8).”

And supplementary methods page 4, lines 1-2:

“In the HFPGR validation cohort, blood samples were collected at enrolment, immediately aliquoted and stored at -70°C for a median of 5.7 years (Q1 4.4; Q3 6.6).”

We furthermore discussed the potential effect of this duration on the results from page 17 line 23 until page 18 line 2:

“The storage duration of the blood samples may have affected the stability of the proteins. We expect the influence on our results to be limited, because storage duration was relatively consistent across patients. While protein degradation might have affected the detectability of individuals proteins and the set of differentially expressed proteins, the clustering approach

is likely robust to this variation, because it is based on relative patterns across the proteome. Nevertheless, application of the clustering model in newly diagnoses HF patients deserves further careful consideration.”

2. There is a discrepancy between the Methods section and the supplementary methods regarding the temperature at which the plasma samples were stored. The manuscripts state -80°C (p.7; s. 13), whereas the supplementary methods state -70°C (p.4; s.5). Please clarify which is correct.

Response: We have updated the manuscript to explain that these two values refer to plasma storage temperatures of the derivation (BioSHiFT) and validation (HFPGR) cohort. Please see page 8, lines 12-13:

“Blood samples of the BioSHiFT cohort were collected at baseline, processed within two hours after collection, and stored at -80°C for a median of 5.3 years (Q1 4.1; Q3 6.8).”

And the supplementary methods page 4, lines 1-2:

“In the HFPGR validation cohort, blood samples were collected at enrolment, immediately aliquoted and stored at -70°C for a median of 5.7 years (Q1 4.4; Q3 6.6).”

3. Also in the supplementary methods, the description (p.4; s.16) alludes to additional quality checks and filtering. For transparency and reproducibility, the authors should describe these processes in more detail, including the filtering criteria applied.

Response: We have updated the manuscript to reflect that QC was not performed specifically for this study but performed and described previously. We also rephrased the sentence to clarify that this step was only processing the aforementioned steps.

Please see supplementary methods, page 4, lines 4-14:

“The quality control was described previously^{2,3}, but in brief, systematic biases in raw assay data were corrected following SomaLogic data standardization protocols, involving multiple normalization and calibration steps. These included Hybridization Control Normalization, Intraplate Median Signal Normalization, and Median Signal Normalization to a global reference. Global reference standards were set for serum and plasma matrices, with controls, QC samples, and calibrators on each plate adjusted to these references. Any deviations in assay performance were monitored over time. An overall protein measurement quality metric was calculated for each sample, with all passing the recommended thresholds. The proteomics data was exported as SomaLogic ADAT files, which were imported into R using the readat package to remove these proteins with low quality.”

This includes the following references:

2. Williams SA, Kivimaki M, Langenberg C, Hingorani AD, Casas JP, Bouchard C, Jonasson C, Sarzynski MA, Shipley MJ, Alexander L, et al. Plasma protein patterns as comprehensive indicators of health. *Nat. Med.* 2019;25:1851–1857.
3. Chadwick J, Hinterberg MA, Asselbergs FW, Biegel H, Boersma E, Cappola TP, Chirinos JA, Coresh J, Ganz P, Gordon DA, et al. Harnessing the Plasma Proteome to Predict Mortality in Heart Failure Subpopulations. *Circ. Heart Fail.* 2025;18:e011208.

4. Similarly, on p.8, s.12-13, the authors mention that correction for the number of independent tests was performed. It is important to specify which method was used for correction for multiple testing and please justify the choice.

Response: Please see page 10, lines 3-8:

“To facilitate model explainability, we identified the differentially expressed proteins between the three proteomics-based clusters using the limma package in R. We corrected for multiple testing using the Benjamini-Hochberg procedure to control the false discovery rate while maintaining a balance between sensitivity and specificity. Proteins were considered significantly differentially expressed if they had an Benjamini-Hochberg adjusted p-value < 0.05 and a minimum absolute log₂-fold-change of 1.0.”

5. The authors should also clarify how missing values were handled for the proteomic data, and whether the proteomic analyses were blinded to the clinical outcome.

Response: The proteomic data did not contain any missing values and the proteomic analyses were indeed blinded to the outcome, which information we have now included in the manuscript on page 8, lines 21-23:

*“Details on the processing and quality control of proteomics data for the validation cohort are provided in the **Supplemental Methods** and the clinical event committee that adjudicated the outcomes was blinded to the proteomic results.”*

6. While the study is validated in a secondary cohort, the reported c-statistics remain modest, with values in the range of 0.69-0.74 for the derivation cohort and 0.65-0.71 for the validation cohort. Such performance is generally considered of limited clinical utility. Although the benchmarking show improvement over clinical characteristic alone, the clinical relevance

of the findings still eludes me. I will encourage the authors to discuss these limitations more explicitly and consider the added benefit for translation into clinical practice.

Response: Please see page 17, lines 1-9 for the suggested discussion on clinical relevance: *“Hence, our results do not show and absence of differences between patients clustered using clinical characteristics or clinical characteristics combined with proteins but rather suggest that the difference is larger when using proteomics exclusively. Despite the improvement compared to clinical clustering, the c-statistics and high dimensional nature of the proteomic clustering limits immediate clinical applicability. Our findings represent a first step toward identifying biologically distinct HFrEF subgroups. Future work should aim to develop a minimal, robust set of proteins that can more accurately assign patients to clusters. These steps were not pursued in the present study to avoid overfitting due to repeated analyses on the same data.”*

Referee #3

This study showed that Proteomics-based HFrEF clustering identified three clusters associated with distinct disease outcomes, which were not detected using clinical characteristics. However, the statistical analysis and related Figures (Tables) seems insufficient in main text. The methods description such as the QC and important variables were not described in detail. Despite the author constructed a validated clustering algorithm, considering the SomaScan proteomic include more than 4000 proteins, it is hard to replicate the clustering due to the cost. The author conducted the differential expression analysis to identify 12 markers, however there is no further analysis to test the accuracy and efficiency of using these 12 markers for phenotyping, nor conducting any other approaches for interpretation and application as sensitivity analysis. These limitations may influence the explanation and clinical implication of this study.

1. The definition of “major cardiovascular (CV) events” should be illustrated in Abstract.

Response: We have included the definition of major CV events in the abstract (page 3, lines 8-11):

“Survival analysis assessed associations with major cardiovascular (CV) events (a composite of HF hospitalization, CV death, heart transplantation, and left ventricular assist device implantation), HF hospitalization, CV death, and all-cause mortality.”

2. I recommend the application interface address can be described in the “Results” section but not in the Abstract and Introduction.

Response: This has been moved as suggested.

3. “Clustering on clinical characteristics identified three patient clusters that did not differ in disease progression rates”. What does the clinical characteristics contain? These information about the input variables should be included in Methods section.

Response: We have now included the list of clinical characteristics in the Methods section as well, which now reads (page 9, lines 2-13):

*“For clinical clustering, variables with more than 5% missingness, related variables, and non-baseline variables arising during follow-up (such as drug prescriptions or devices) were excluded. Continuous clinical variables age, body mass index, and mean arterial pressure were categorized based on clinically relevant cut-offs, such as the World Health Organization classifications of normal weight, overweight, and obese for body mass index (Table S3), and for the plasma proteins, the principal components (PCs) were categorized using quartiles. Identification of the optimal number of clusters and categorization is described in the **Supplemental Methods** and **Figure S1**. A genetic algorithm was employed to identify the subset of clinical variables used in LCA. In short, the genetic algorithm performs an exhaustive search for the most relevant clustering variables, comparing multiple subsets of variables simultaneously and by slightly mutating these subsets over many iterations^{14,15}.”*

And on page 9, lines 20-22:

“We next determined the contrasts between HFrEF clusters based on clinical characteristics (age, aetiology, and history of coronary artery disease [CAD], arrhythmia, hypertension, and smoking) and clusters based on the first 20 PCs of the plasma proteins.”

4. The Quality Control Process and Results of Quality (such as important QC statistics and QC Figures) should be included in main text (It is better to demonstrate a summary of QC in main text and to illustrate in detail in the Supplementary Material).

Response: The quality control of the proteomics was not performed specifically for this study and has previously been described in more detail. We have mentioned this and included references to these studies on page 8, lines 14-16:

“Proteomic analyses of EDTA plasma samples were performed using the aptamer-based proteomic SOMAscan platform¹² and the quality control of the proteomics has been described previously^{8,13}.”

This includes the following references:

8. Petersen TB, de Bakker M, Asselbergs FW, Harakalova M, Akkerhuis KM, Brugts JJ, van Ramshorst J, Lumbers RT, Ostroff RM, Katsikis PD, et al. HF_rEF subphenotypes based on 4210 repeatedly measured circulating proteins are driven by different biological mechanisms. *EBioMedicine*. 2023;93.
12. Gold L, Ayers D, Bertino J, Bock C, Bock A, Brody E, Carter J, Cunningham V, Dalby A, Eaton B, et al. Aptamer-based multiplexed proteomic technology for biomarker discovery. *Nat. Preced.* 2010;1–1.
13. Williams SA, Kivimaki M, Langenberg C, Hingorani AD, Casas JP, Bouchard C, Jonasson C, Sarzynski MA, Shipley MJ, Alexander L, et al. Plasma protein patterns as comprehensive indicators of health. *Nat. Med.* 2019;25:1851–1857.

And in the supplemental methods, page 4, lines 4-6:

“The quality control was described previously^{2,3}, but in brief, systematic biases in raw assay data were corrected following SomaLogic data standardization protocols, involving multiple normalization and calibration steps.”

This includes the following references:

2. Williams SA, Kivimaki M, Langenberg C, Hingorani AD, Casas JP, Bouchard C, Jonasson C, Sarzynski MA, Shipley MJ, Alexander L, et al. Plasma protein patterns as comprehensive indicators of health. *Nat. Med.* 2019;25:1851–1857.
3. Chadwick J, Hinterberg MA, Asselbergs FW, Biegel H, Boersma E, Cappola TP, Chirinos JA, Coresh J, Ganz P, Gordon DA, et al. Harnessing the Plasma Proteome to Predict Mortality in Heart Failure Subpopulations. *Circ. Heart Fail.* 2025;18:e011208.

5. “A sensitivity analysis based on the first 30 protein PCs similarly resulted in three clusters with comparable outcome associations.”. How are the first 30 proteins determined?

Response: For the proteomic clustering, we used the principal components (PCs) of the protein measurements to reduce dimensionality. In the main analysis we included the first 20 PCs as clustering variables. For the sensitivity analysis we included an additional 10 PCs to

increase the variance explained in these PCs. We have clarified this in the manuscript on page 9 lines 9-17:

*“A genetic algorithm was employed to identify the subset of clinical variables used in LCA. In short, the genetic algorithm performs an exhaustive search for the most relevant clustering variables, comparing multiple subsets of variables simultaneously and by slightly mutating these subsets over many iterations^{14,15}. For the clustering on plasma protein levels, the first 20 PCs were used, and a sensitivity analysis was performed using the first 30 PCs (**Figure S1**). Additionally, we combined the clinical clustering variables with the first 20 PCs to evaluate the potential improvement in clustering performance from integrating these features (**Supplemental Methods**).”*

6. How does the statistical significance of differential expressed proteins test? Such as the cut-off of P-value? And is the comparison conducted in 3 clusters to report P for trend, or other comparison methods?

Response: We have included the following text in the manuscript to clarify the statistical procedure for identifying the differentially expressed proteins (page 10, lines 3-8):

“To facilitate model explainability, we identified the differentially expressed proteins between the three proteomics-based clusters using the limma package in R. We corrected for multiple testing using the Benjamini-Hochberg procedure to control the false discovery rate while maintaining a balance between sensitivity and specificity. Proteins were considered significantly differentially expressed if they had an Benjamini-Hochberg adjusted p-value < 0.05 and a minimum absolute log₂-fold-change of 1.0.”

7. The Figure Legend and Figure explanation of Figure 1 were missing.

Response: The figure legend and explanation are supplied on page 22:

“Figure 1. HFrEF patient flow between the different clustering models.

N.B. The flow represents the number of HFrEF patients that were assigned to different clusters in the various approaches. For example, this figure shows that the clinical clusters (i.e. ischemic cluster 1, hypertensive cluster 2, and cardiomyopathy cluster 3) are very different from the proteomic clusters, whereas the proteomic slowly progressing cluster 1 and combined cluster 1 are almost identical.”

8. The description for characteristics of each cluster should be better demonstrate in both text and figure (table). I recommend a summary of each cluster’s characteristic should also be included in Abstract.

Response: We have now included a figure to display the proportion of clinical characteristics per cluster (**Figure R2** below, which is included in the manuscript as **Figure S4**) and a summary of the proteomics clusters in the abstract on page 3, lines 14-18:

“Proteomics-based clustering identified three clusters associated with disease progression. Cluster 1 included younger patients with fewer comorbidities, whereas cluster 3 consisted of older patients with more atrial fibrillation and renal failure. Cluster 2 had intermediate values for most characteristics; medication use was similar across clusters.”

Figure R2. Proportion of clinical characteristics per cluster

Abbreviations: AF = atrial fibrillation, CAD = coronary artery disease, MI = myocardial infarction, NYHA = New York Heart Association.

Referee #4

This manuscript describes the use of proteomics data generated using SomaScan to discover latent classes. The main aim is to identify plasma protein profiles of HF patients with reduced ejection fraction (HFrEF).

1. The choice to discretize PCA scores into quartiles seems arbitrary. It would be helpful for the authors to justify why quartiles were selected, and to clarify whether different categorization schemes were evaluated for robustness (i.e., whether the final clustering results depend heavily on how the PCA scores were divided).

Response: We have elaborated on this in the supplementary methods section, where we explain that we used entropy to define the optimal choice for categorizing the PCs.

We refer to the supplements in the methods section on page 9, lines 8-9:

*“Identification of the optimal number of clusters and categorization is described in the **Supplemental Methods and Figure S1.**”*

And in the supplements from page 2 line 25 until page 3 line 2:

“Categorization of the principal components was done using different percentile-based cut-offs: quartiles (25th, 50th, 75th), deciles (10% intervals), as well as custom groupings based on the 10th–90th and 20th–80th percentiles to define low, middle, and high values. Clustering was performed using all types of categorization and the optimal categorization was chosen based on the highest entropy.”

2. It’s not very clear whether the latent classes reflect actual patient subgroups or are potentially artifacts of preprocessing choices due to the use of PCA to categorize the proteomics data.

Response: We fully agree with the reviewer’s voiced concern. To address this, we have externally validated the derived clustering model in a completely independent cohort. This confirms the presented results, and the derived algorithm is not driven by pre-processing artifacts.

This has been discussed in the limitations section as follows (page 18, lines 8-13):

“Due to the moderate sample size, we categorised continuous variables to simplify the model and reduce sample size requirements, which might have affected model expressivity. In larger sample size settings, more flexible algorithms may further improve the already highly discriminative model’s current performance. Nevertheless, the derived model was successfully validated in an external cohort that differed considerably from the derivation cohort, underscoring the robustness of our approach.”

3. Did the authors consider using LPA instead of LCA for the proteomics data?

Response: We thank the reviewer for this suggestion which has been implemented as an additional sensitivity analysis.

Please see page 9, lines 17-18:

*“Another sensitivity analysis using latent profile analysis is described in the **Supplemental Methods**.”*

And the results on page 13, lines 1-3:

“A sensitivity analysis based on the first 30 protein PCs similarly resulted in three clusters with comparable outcome associations and latent profile analysis did not identify clusters with a similarly strong association with disease progression.”

For the relevant methods section, please see appendix page 3, lines 12-14:

“As a sensitivity analysis we performed the proteomic clustering using latent profile analysis (LPA) using the R package mclust, which does not require categorization of the continuous principal components.”

Appendix results on page 5 line 25 until page 6 line 4:

*“The sensitivity analysis using LPA resulted in three clusters, with the first cluster consisting of younger individuals with a median age of 64 years (Q1 55; Q3 71) and the highest percentage of females (32%). The second cluster grouped 17 patients, with a median age of 72 years (Q1 68; Q3 74), 12 patients suffering from renal failure (71%) and six from diabetes (35%). The 125 patients in the third cluster had a median age of 65 years (Q1 53; Q3 74) and had the highest New York Heart Association class (37% III/IV). Strikingly, the LVEF was equal among the three clusters (**Appendix Table S13**).”*

Appendix Table S13. Baseline table of the Bio-SHiFT HFrEF patients stratified by proteomics cluster using LPA

	Cluster 1 (n = 237)	Cluster 2 (n = 17)	Cluster 3 (n = 125)	p*
n	237	17	125	-
Demographics	-	-	-	-
Male sex, n (%)	162 (68.4)	13 (76.5)	100 (80.0)	0.058
Age (years) (median [IQR])	64.00 [55.00, 71.00]	72.00 [68.00, 74.00]	65.00 [53.00, 74.00]	0.021
European ethnicity, n (%)	220 (93.6)	15 (88.2)	113 (91.1)	-
Risk factors	-	-	-	-
BMI (kg/m²) (median [IQR])	26.51 [24.11, 30.33]	27.92 [24.93, 29.34]	26.05 [23.52, 29.10]	0.224
Systolic blood pressure (mmHg) (median [IQR])	118.00 [100.00, 130.00]	110.00 [107.00, 120.00]	110.00 [100.00, 126.00]	0.227
Diastolic blood pressure (mmHg) (median [IQR])	70.00 [63.00, 78.00]	70.00 [60.00, 78.00]	70.00 [60.00, 80.00]	0.201
Mean arterial pressure (mmHg) (median [IQR])	86.67 [76.67, 94.33]	83.33 [76.67, 90.00]	83.33 [73.33, 93.00]	0.194
Hypertension, n (%)	99 (42.3)	9 (52.9)	56 (45.2)	0.643
Hypercholesterolemia, n (%)	104 (44.6)	9 (56.2)	46 (38.0)	0.269
Smoking, n (%)	173 (73.3)	11 (64.7)	85 (68.5)	0.526
HF-related measures	-	-	-	-
Etiology, n (%)	-	-	-	0.633
 Cardiomyopathy	76 (32.1)	4 (23.5)	43 (34.4)	-
 Exposure toxic substances	13 (5.5)	2 (11.8)	5 (4.0)	-
 Hypertension	17 (7.2)	4 (23.5)	12 (9.6)	-
 Infiltrative disease	1 (0.4)	0 (0.0)	1 (0.8)	-
 Ischemic heart disease	105 (44.3)	7 (41.2)	51 (40.8)	-

Myocarditis	3 (1.3)	0 (0.0)	0 (0.0)	-
Unknown	15 (6.3)	0 (0.0)	8 (6.4)	-
Valvular disease	7 (3.0)	0 (0.0)	5 (4.0)	-
NYHA class = III/IV, n (%)	53 (22.5)	5 (29.4)	46 (37.1)	0.013
LVEF (%) (median [IQR])	30.00 [23.00, 38.00]	30.00 [23.00, 35.00]	30.00 [20.00, 36.00]	0.676
Clinical history	-	-	-	-
CAD, n (%)	103 (43.6)	7 (41.2)	55 (44.0)	0.566
MI, n (%)	96 (41.2)	7 (43.8)	41 (32.8)	0.270
Valvular heart disease, n (%)	132 (55.9)	9 (56.2)	71 (57.3)	0.971
AF, n (%)	78 (33.5)	9 (52.9)	48 (38.7)	0.207
Arrhythmia, n (%)	91 (38.9)	4 (23.5)	55 (44.0)	0.237
Stroke, n (%)	26 (11.1)	2 (12.5)	20 (16.1)	0.401
Renal failure, n (%)	100 (42.2)	12 (70.6)	67 (54.0)	0.015
Diabetes, n (%)	56 (23.6)	6 (35.3)	35 (28.0)	0.427
PCI, n (%)	86 (36.3)	5 (29.4)	34 (27.2)	0.206
CABG, n (%)	31 (13.1)	4 (23.5)	18 (14.4)	0.480
Pacemaker, n (%)	49 (21.4)	2 (13.3)	32 (26.2)	0.399
ICD, n (%)	162 (68.4)	10 (58.8)	80 (64.0)	0.558
CRT, n (%)	73 (30.8)	5 (29.4)	33 (26.6)	0.709
Biomarkers	-	-	-	-
Creatinine (mg/dL) (median [IQR])	1.21 [1.00, 1.49]	1.33 [1.01, 1.94]	1.26 [1.03, 1.52]	0.279
Cystatin C (mg/L) (median [IQR])	0.70 [0.54, 0.88]	0.87 [0.74, 0.99]	0.79 [0.61, 1.01]	0.026

Hs-Troponin T (ng/L) (median [IQR])	15.00 [9.14, 28.00]	15.19 [13.57, 32.05]	26.36 [17.04, 45.36]	<0.001
CRP (mg/L) (median [IQR])	2.00 [0.90, 4.30]	1.60 [0.95, 6.40]	2.50 [1.00, 5.75]	0.368
eGFR (mL/min/1.73m²) (median [IQR])	58.28 [44.21, 76.03]	50.23 [29.44, 67.60]	56.03 [43.91, 72.61]	0.287
Medication	-	-	-	-
Beta blockers, n (%)	222 (93.7)	17 (100.0)	109 (87.2)	0.046
ACE inhibitors, n (%)	160 (67.5)	9 (52.9)	87 (69.6)	0.388
Angiotensin II receptor blockers, n (%)	71 (30.0)	5 (29.4)	30 (24.0)	0.482
Aldosteron antagonists, n (%)	188 (79.3)	11 (64.7)	91 (72.8)	0.190
Digoxin, n (%)	76 (32.1)	8 (47.1)	44 (35.2)	0.414
Anticoagulantia, n (%)	170 (71.7)	14 (82.4)	92 (73.6)	0.618
Statins, n (%)	139 (58.6)	11 (64.7)	63 (50.4)	0.248
Diabetic control, n (%)	53 (22.4)	6 (35.3)	31 (24.8)	0.454
Outcomes	-	-	-	-
Days to censoring (median [IQR])	828.00 [509.00, 955.00]	926.00 [790.00, 1,065.00]	781.00 [432.00, 928.00]	0.056
HF Hospitalization, n (%)	43 (18.1)	4 (23.5)	42 (33.6)	0.004
Days to HF hospitalization (median [IQR])	782.00 [445.00, 944.00]	923.00 [553.00, 1,065.00]	633.00 [363.00, 891.00]	0.011
Cardiovascular death, n (%)	13 (5.5)	3 (17.6)	16 (12.8)	0.022
Days to cardiovascular death (median [IQR])	828.00 [518.00, 955.00]	926.00 [790.00, 1,065.00]	781.00 [432.00, 928.00]	0.046
All-cause mortality, n (%)	19 (8.0)	3 (17.6)	19 (15.2)	0.073
Days to all-cause mortality (median [IQR])	835.00 [560.00, 956.00]	926.00 [790.00, 1,065.00]	781.00 [436.00, 928.00]	0.038

** Continuous variables were compared with the Kruskal-Wallis test and categorical variables were compared with the Chi-square test.*

Abbreviations: ACE = Angiotensin-Converting Enzyme, AF = atrial fibrillation, BMI = body mass index, CABG = coronary artery bypass grafting, CAD = coronary artery disease, CRP = c-reactive protein, CRT = cardiac resynchronization therapy, eGFR = estimated Glomerular Filtration Rate, HF = heart failure, ICD = implantable cardioverter-defibrillator, IQR = interquartile range, LVEF = left ventricular ejection fraction, MI = myocardial infarction, NT-proBNP = N-terminal pro-B-type natriuretic peptide, NYHA = New York Heart Association, PCI = percutaneous coronary intervention

And on page 6, lines 14-17:

“Taking cluster 1 identified by the LPA analysis as a reference, patients in cluster 3 had an increased event rate for all outcomes, with limited difference comparing outcomes of cluster 2 to cluster 1. The c-statistic ranged between 0.58 and 0.63 for all outcomes (**Appendix Table S14**).”

Appendix Table S14. Difference in survival of patients for proteomic clustering using LPA

Outcome	Cluster	HR (95% CI)	p-value	Overall p-value	c-statistic (95% CI)
Major CV event	Cluster 1	Reference	-	0.007	0.58 (0.53; 0.63)
	Cluster 2	1.09 (0.43; 2.71)	0.860		
	Cluster 3	1.86 (1.27; 2.71)	0.001		
HF hospitalization	Cluster 1	Reference	-	0.003	0.6 (0.54; 0.65)
	Cluster 2	1.16 (0.41; 3.24)	0.779		
	Cluster 3	2.10 (1.37; 3.22)	6.13×10 ⁻⁴		
CV death	Cluster 1	Reference	-	0.026	0.63 (0.54; 0.71)
	Cluster 2	3.00 (0.85; 10.54)	0.086		
	Cluster 3	2.54 (1.22; 5.27)	0.013		
All-cause mortality	Cluster 1	Reference	-	0.075	0.59 (0.51; 0.67)
	Cluster 2	1.95 (0.57; 6.61)	0.284		
	Cluster 3	2.06 (1.09; 3.89)	0.026		
Hazard ratios and confidence intervals were calculated using a univariable cox regression. Follow-up was truncated at three years. Abbreviations: CI = confidence interval, CV = cardiovascular, HR = hazard ratio					

4. Have the authors consider using pathway enrichment or gene set enrichment analysis (e.g., GSEA) within each cluster to better understand the molecular processes or pathways underlying the observed clusters?

Response: The reviewer provided a meaningful suggestion and we have now performed an enrichment analysis using the Gene Ontology (GO) resource. Please see the updated methods section on page 10, lines 6-12:

“Proteins were considered significantly differentially expressed if they had an Benjamini-Hochberg adjusted p-value < 0.05 and a minimum absolute log₂-fold-change of 1.0. Each protein was assigned to the cluster with the highest expression. To gain biological insight, we next explored pathway enrichment using the R package clusterProfiler, leveraging the gene ontology resource and applying a Benjamini-Hochberg adjusted p-value threshold of 0.05 on proteins with a minimum absolute log₂-fold change of 0.7.”

And the results section on page 13 line 26 until page 14 line 5:

*“Filtering for a log₂-fold-change of 0.7 resulted in 56 additional proteins and pathway enrichment analysis showed that slowly progressing cluster 1 was characterised by proteins involved in lipid metabolism (such as ADH4, APOA5, APOC3, and CRYZL1). Rapidly progressing cluster 3 was characterised by proteins involved in the immune response (such as EPHA2, RNase1, MMP12, NBL1, and REG-3-alpha) and extracellular matrix organisation (such as COL28A1, CST3, PRSS2, PXDN, and TNFRSF1A). Hospitalization cluster 2 was characterised by energy metabolism and glucose usage (**Table S11**).”*

In the discussion, these results were integrated on page 15, lines 11-18:

“Out of the twelve differentially expressed proteins, eight have been previously associated with cardiac traits or diseases. Specifically, common genetic variants in the ABO gene have been associated with heart failure¹⁸⁻²¹ and myocardial infarction²²⁻²⁴, while variants in TP53I11 have been associated with resting heart rate^{25,26} and hypertrophic cardiomyopathy²⁷. These proteins were significantly upregulated in slowly progressing cluster 1, which was characterised by an altered lipid homeostasis. Hospitalization cluster 2 was intermediate in many aspects, such as expression of the proteins and characterised by glucose usage in the pathway analysis.”

And on page 15 line 24 until page 16 line 4:

“RNase1, NBL1, and REG-3-alpha are involved in immune response mechanisms and together with PXDN, these proteins and processes characterise the rapidly progressing cluster 3. Both IGFBP-1 and IGFBP-2 are involved in inflammation and metabolism, and

modulate the effects of IGF-1, which is crucial for cellular development and survival. They are hypothesised to play a role in HF pathophysiology and have been suggested as biomarkers for mortality and CV disease risk^{30–33}. The combination of high levels of IGFBP-1 and NT-proBNP have previously been associated with worse prognosis in HFrEF patients⁷.”

5. While using \log_2 fold change >1 is reasonable, the authors may consider exploring a lower cutoff (e.g., 0.5) to capture more subtle shifts between clusters.

Response: We thank the reviewer for this suggestion and have extended the supplementary table on the significant differentially expressed proteins to include the proteins meeting a lower cutoff of 0.7. This approach captures 56 additional, potentially meaningful proteins, which we used to perform the newly included GO enrichment analysis on.

We have reported this in the manuscript as follows on page 10, lines 3-12:

“To facilitate model explainability, we identified the differentially expressed proteins between the three proteomics-based clusters using the limma package in R. We corrected for multiple testing using the Benjamini-Hochberg procedure to control the false discovery rate while maintaining a balance between sensitivity and specificity. Proteins were considered significantly differentially expressed if they had an Benjamini-Hochberg adjusted p-value < 0.05 and a minimum absolute \log_2 -fold-change of 1.0. Each protein was assigned to the cluster with the highest expression. To gain biological insight, we next explored pathway enrichment using the R package clusterProfiler, leveraging the gene ontology resource and applying a Benjamini-Hochberg adjusted p-value threshold of 0.05 on proteins with a minimum absolute \log_2 -fold change of 0.7.”

And from page 13 line 18 until page 14 line 5:

“Next, to improve model interpretability, we identified differentially expressed proteins across the three proteomics-based derived clusters. We observed that ABO and tumor protein p53-inducible protein 11 (TP53I11) values were significantly upregulated in the slowly progressing cluster 1, whereas the values of ten proteins were significantly upregulated in the rapidly progressing cluster 3 (showing intermediate high values in hospitalization cluster 2): peroxidase homolog (PXDN), neuroblastoma suppressor of tumorigenicity 1 (NBL1), RNase1, insulin-like growth factor binding protein (IGFBP) 2, REG-3-alpha, growth-differentiation factor-15 (GDF-15), Apolipoprotein F, NT-proBNP, IGFBP-1, and CD2 (Table S10). Filtering for a \log_2 -fold-change of 0.7 resulted in 56 additional proteins and pathway enrichment analysis showed that slowly progressing cluster 1 was characterised by proteins involved in lipid metabolism (such as ADH4, APOA5, APOC3, and CRYZL1). Rapidly

progressing cluster 3 was characterised by proteins involved in the immune response (such as EPHA2, RNase1, MMP12, NBL1, and REG-3-alpha) and extracellular matrix organisation (such as COL28A1, CST3, PRSS2, PXDN, and TNFRSF1A). Hospitalization cluster 2 was characterised by energy metabolism and glucose usage (Table S11)."

Response to the reviewers

Response: We thank the reviewers for their thoughtful comments and constructive feedback, which have greatly improved the quality of our manuscript.

Reviewer #1

The author answered my comments well and discussed the limitations properly. I had no further comments. Thank you.

Reviewer #2

I have no further comments to the manuscript. The authors have adequately addressed my previous comments.

Reviewer #3

The authors have addressed and revised most of my comments. I still have some minor suggestions regarding the tables and figures in the manuscript. If there is no restriction on the number of tables and figures, I would still recommend increasing their quantity—for example, by moving Figure S4 and S5 into the main text.

Response: We have now moved the figures to the main text, they are included as Figure 1 and Figure 5.

As the authors highlighted in the abstract: "Twelve proteins were differentially expressed, including druggable targets CD2, GDF-15, ABO, IGFBP-1, IGFBP-2, and RNase1." It is also advisable to add a table listing the drug-target profiles of these key proteins.

Response: We have included a reduced version of Supplementary Table 14 in the main text as Table 2.

Reviewer #4

Many thanks to the authors for their response. I have no further comments.

Response to the editors

Response: We thank the editors for their clarifying remarks, which have improved the quality of our manuscript.

1. Remove tables 1 and 2 from the main manuscript file (as they are provided as Supplementary Data files).

We have removed the tables and their references from the manuscript.

2. In the Methods part of the Abstract, explain druggability as this term may not be understood by all readers.

We have adapted the following sentence to explain druggability: We identified differentially expressed proteins and explored whether proteins are targets of developmental or approved drugs.

3. In the Methods section, please add a statement about written consent from participants in the Bio-SHiFT study.

We have included the following statement on the informed consent: All patients provided written informed consent.

4. For Supplementary Figures 2-3, the figure titles must describe the figure as a whole and must not contain reference to specific figure panels. The titles should be brief. All figure panels must be defined in the legend.

We have changed the figure titles and legends to the following (underlined text is changed):

Supplementary Figure 2. Criteria for choosing the number of clusters.

The optimal number of clusters was defined using the Bayesian Information Criterion (BIC), depicted for A) clinical, C) proteomic, and E) combined clustering, and the log likelihood ratio distribution (BLRT), depicted for B) clinical, D) proteomic, and F) combined clustering. We compared the k cluster model with the lowest BIC with a

k+1 cluster model calculating the likelihood ratio for the two models. The distribution of the likelihood ratio was estimated using 999 bootstraps, selecting the model with $k+1$ clusters based on a p -value of 0.05 or smaller. Vertical lines represent the observed log likelihood difference with which the p -values were determined. The p value was obtained by comparing the observed difference to this distribution, using a one-sided test and no multiple testing correction.

Supplementary Figure 3. Event-free survival per clinical outcome for the clusters.

Kaplan-Meier curve for the clinical outcomes stratified by A) clinical and B) combined clusters. Differences were assessed using the log-rank test. Abbreviations: HF = heart failure.

5. Please indicate the statistical test used for data analysis in the legend(s) of Table s8

The following text was included in the legend of Table S8:

Hazard ratios and confidence intervals were calculated using a univariable cox regression. Follow-up was truncated at three years. Tests were two-sided and not adjusted for multiple testing.

6. Please note that the exact p value should be provided, when possible, in the legend(s) of Tables S4, S5, S6.

Where possible, the exact p -values were provided.

7. If data are reported in the study, the statement should specify, at a minimum, that all relevant data are available from the authors upon request. This statement should include details on who will be responsible for replying to this request and their contact details (email address).

The following statement was included to allow for data requests:

Anonymized data that support the findings of this study will be made available to other researchers for purposes of reproducing the results upon reasonable request

and in accordance with a data-sharing agreement. Requests can be directed to I. Kardys (i.kardys@erasmusmc.nl).

8. Please add a sentence to the Data Availability statement to describe how source data can be accessed using text similar to “The source data for Figure X is in Supplementary Data Y”. It needs to be specific about which figures you provide source data for, and specifically where it can be found.

We have included an additional Supplementary Table, to provide the underlying data for Figure 5. To be more specific about the underlying data, we have included the following text to the data availability statement:

The source data for Figure 1 is in Supplementary Table 4, for Figure 3 and Supplementary Figure 3 in Supplementary Table 8, for Figure 4 in Supplementary Table 10, and for Figure 5 in Supplementary Table 14.